# Beyond accuracy: quantifying trial-by-trial behaviour of CNNs and humans by measuring error consistency

**Robert Geirhos**[*]
University of Tübingen & IMPRS-IS
robert.geirhos@uni-tuebingen.de

**Kristof Meding**[*]
University of Tübingen
kristof.meding@uni-tuebingen.de

**Felix A. Wichmann**
University of Tübingen
felix.wichmann@uni-tuebingen.de

[*] Joint first authors (alphabetical order)

## Abstract

A central problem in cognitive science and behavioural neuroscience as well as in machine learning and artificial intelligence research is to ascertain whether two or more decision makers—be they brains or algorithms—use the same strategy. Accuracy alone cannot distinguish between strategies: two systems may achieve similar accuracy with very different strategies. The need to differentiate beyond accuracy is particularly pressing if two systems are at or near ceiling performance, like Convolutional Neural Networks (CNNs) and humans on visual object recognition. Here we introduce trial-by-trial *error consistency*, a quantitative analysis for measuring whether two decision making systems systematically make errors on the same inputs. Making consistent errors on a trial-by-trial basis is a necessary condition if we want to ascertain similar processing strategies between decision makers. Our analysis is applicable to compare algorithms with algorithms, humans with humans, and algorithms with humans.

When applying error consistency to visual object recognition we obtain three main findings: (1.) Irrespective of architecture, CNNs are remarkably consistent with one another. (2.) The consistency between CNNs and human observers, however, is little above what can be expected by chance alone—indicating that humans and CNNs are likely implementing very different strategies. (3.) CORnet-S, a recurrent model termed the "current best model of the primate ventral visual stream", fails to capture essential characteristics of human behavioural data and behaves essentially like a standard purely feedforward ResNet-50 in our analysis; highlighting that certain behavioural failure cases are not limited to feedforward models. Taken together, error consistency analysis suggests that the strategies used by human and machine vision are still very different—but we envision our general-purpose error consistency analysis to serve as a fruitful tool for quantifying future progress.

## 1 Introduction[1]

Complex systems are notoriously difficult to understand—be they Convolutional Neural Networks (CNNs) or the human mind or brain. Paradoxically, for CNNs, we have access to every single model parameter, know exactly how the architecture is formed of stacked convolution layers, and

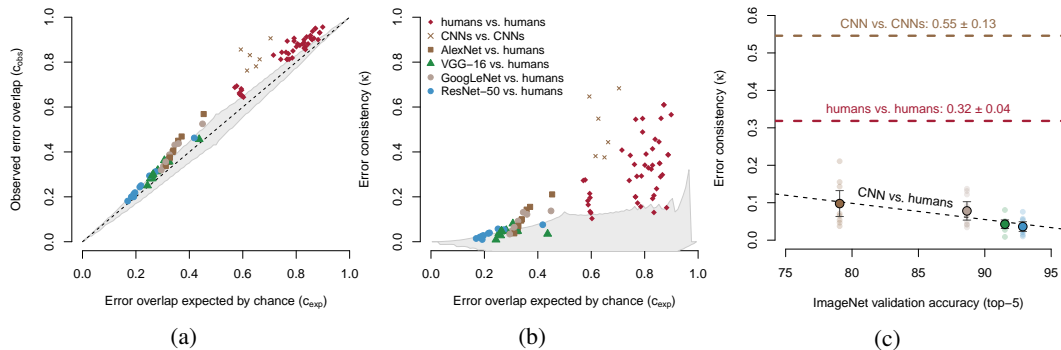

Figure 1: Do humans and CNNs make consistent errors? From left to right three steps for analysing this question are visualised. For a detailed description of these steps please see the intuition (1.1). **(a)** Observed vs. expected error overlap (errors on the same trials) for a classification experiment where humans and CNNs classified the same images [11]. Values above the diagonal indicate more overlap than expected by chance. **(b)** Same data as on the left but measured by error consistency ($\kappa$). Higher values indicate greater consistency; shaded areas correspond to a simulated 95% percentile for chance-level consistency. **(c)** Error consistency vs. ImageNet accuracy.

we can inspect every single pixel of the training data—yet understanding the behaviour emerging from these primitives has proven surprisingly challenging [1], leaving us continually struggling to reconcile the success story of CNNs with their brittleness [2–4].[2] In response to the need to better understand the internal mechanisms, a number of visualisation methods have been developed [6–8]. And while many of them have proven helpful in fuelling intuitions, some have later been found to be misleading [9, 10]; moreover, most visualisation analyses are qualitative at nature. On the other hand, quantitative comparisons of different algorithms like benchmarking model accuracies have led to a lot of progress across deep learning, but reveal little about the internal mechanism: two models may reach similar levels of accuracy with very different internal processing strategies, an aspect that is gaining importance as CNNs are rapidly approaching ceiling performance across tasks and datasets. In order to understand whether two algorithms are implementing a similar or a different strategy, we need analyses that are quantitative *and* allow for drawing conclusions about the internal mechanism.

We here introduce *error consistency*[3], a quantitative analysis for measuring whether two black-box perceptual systems systematically make errors on the same inputs. Irrespective of any potential differences at Marr's implementational level [12] (which may be quite large, e.g. between two different neural network architectures or even larger between a CNN and a human observer), one can only conclude that two systems use a similar strategy if these systems make similar errors: not just a similar number of errors (as measured by accuracy), but also errors on the same inputs, i.e. if two systems find the same *individual* stimuli difficult or easy (as measured by error consistency). An agreement can be considered inverse to the Reichenbach-principle [13] of correlation: correlation between variables does not imply a direct causal relationship. However, correlation does imply *at least* an indirect causal link through other variables. For error consistency, zero error consistency implies that two decision makers are not using the same strategy. While error consistency can be applied across fields, tasks and domains (including vision, auditory processing, etc.), we believe it to be of particular relevance at the intersection of deep learning, neuroscience and cognitive science. Both brains and CNNs have, at various points, been described as black-box mechanisms [14–16]. But do the spectacular advances in deep learning shed light on the perceptual and cognitive processes of biological vision? Does similar performance imply similar mechanism or algorithm? Do different CNNs indeed make different errors?[4] We believe that fine-grained analysis techniques like error consistency may serve an important purpose in this debate.

**Molecular psychophysics.** Analysing errors for every single input is inspired by the idea of "molecular psychophysics" by David Green [18]. He argued that the goal of psychophysics should be to predict human responses to individual stimuli (trials) and not only aggregated responses (accuracy), let alone only averages across many individuals, as is common in much of the behavioural sciences. Green also predicted that once models of perceptual processes became more advanced, accuracy would cease to be a good criterion to assess and compare them rigorously (see p. 394 in [18]).

**Related work.** Using error consistency we can analyse human and CNN error patterns in a way that has, we believe, not been done before. We obtain *novel findings* but we do not consider error consistency to be an entirely *novel method* by itself. Instead, it builds on, extends and adapts existing methods and ideas developed in three different fields: molecular psychophysics (as described above) as well as causal inference and the social sciences (as described below). Our goal is the systematic analysis of human and CNN error patterns at the trial-by-trial level. Many previous analyses have focused on the aggregated level instead: In machine learning, performance is predominantly measured by accuracy and existing metrics to analyse errors such as comparisons between confusion matrices [19–23] or scores based on KL divergence [24] pool over single trials, thereby losing crucial information—they are not "molecular" but only "molar" in Green's terminology [18]. [25, 26] went an important step further by comparing errors at an image-by-image level, but consistency was only computed *after* aggregating across participants, and [25] use a metric that automatically leads to higher consistency when comparing two systems with higher accuracy (without discounting for consistency due to chance). Closely related to our analysis is [27], who investigated similarity between models in the context of overfitting. In the context of causal inference, [28] performed a trial-by-trial analysis, plotting expected vs. observed behaviour (a starting point for our analysis). In social sciences, psychology and medicine, comparisons between participants are common, e.g. for problems like "How do people differ when answering a questionnaire?". In that context, so-called inter-rater agreement is measured by Cohen's kappa [29]. Here we repurpose and extend Cohen's kappa ($\kappa$) for the analysis of classification errors by humans and machines, and provide confidence intervals and analytical bounds (limiting possible consistency).

**Terminology.** A *decision maker* is any (living or artificial) entity that implements a decision rule. A *decision rule* is a function that defines a mapping from input to output (see [4] for a taxonomy of decision rules). Note that the same decision rule can result from different strategies. We use the term *strategy* synonymously with the term *algorithm*. For instance, `Quicksort(X)` and `Mergesort(X)` use a different algorithm (strategy), but they implement the same decision rule: the output will always be the same. `Permute(X)`, on the other hand, will (usually) lead to a different output. Hence, similar output (or similar errors, i.e., high error consistency) is a necessary, but not a sufficient condition for similar strategies.

## 1.1 Intuition

Before going through the mathematical details in Section 2, let us consider a simple example of a psychophysical experiment where human observers and CNNs classified objects from 160 images (line drawing / edge-like stimuli in this case). There are three steps in order to analyse error consistency (visualised in Figure 1). We can start by analysing how many of the decisions (either correct or incorrect) to individual trials are identical (*observed error overlap*). This number only becomes meaningful when plotted against the *error overlap expected by chance* (Figure 1a): for instance, two observers with high accuracies will necessarily agree on many trials by chance alone. However, this visualisation may be hard to interpret since higher values do not simply correspond to higher consistency (instead, above-chance consistency is measured by distance from the diagonal). In a second step, we can therefore normalise the data (Figure 1b) by dividing each datapoint's distance to the diagonal by the total distance between the diagonal and ceiling (1.0). Now, we can directly compare the error consistency between decision makers: if error consistency is measured by $\kappa$, then $\kappa = 0$ means chance-level consistency (independent processing strategies), $\kappa > 0$ indicates consistency beyond chance (similar strategies) and $\kappa < 0$ inconsistency beyond chance (inverse strategies). Lastly, we can analyse the relationship between error consistency ($\kappa$) and an arbitrary other variable, for instance in order to determine whether better ImageNet accuracy leads to higher consistency between a CNN and human observers (Figure 1c), which is not the case here.

## 2 Methods

When comparing two decision makers the most obvious comparison is accuracy. Our goal is to go beyond accuracy per se by assessing the consistency of the responses with respect to individual stimuli. As a prerequisite, all decision makers need to evaluate the exact same stimuli. The order of presentation is irrelevant as long as the responses can be sorted w.r.t. stimuli afterwards.[5] In the following, we show how error consistency can be computed and which bounds and confidence intervals apply for the observed error overlap (2.1) and for $\kappa$ (2.2). Experimental methods are described in 2.3 and code is available from `https://github.com/wichmann-lab/error-cons istency`.

### 2.1 Observed vs. expected error overlap

If two observers $i$ and $j$ (be they algorithms, humans or animals) respond to the same $n$ trials, we can investigate by how much their decisions overlap. For this purpose, we only analyse whether the decisions were correct/incorrect (irrespective of the number of choices). The observed error overlap $c_{obs}$ is defined as $c_{obs_{i,j}} = \frac{e_{i,j}}{n}$ where $e_{i,j}$ is the number of equal responses (either both correct or both incorrect). In order to find out whether this observed overlap is beyond what can be expected by chance, we can compare observers $i$ and $j$ to a theoretical model: independent binomial observers (binomial: making either a correct or an incorrect decision; independent: only random consistency). In this case, we can expect only overlap due to chance $c_{exp_{i,j}}$:

$$c_{exp_{i,j}} = p_i p_j + (1 - p_i)(1 - p_j). \tag{1}$$

This is the sum of the probabilities that two observers $i$ and $j$ with accuracies $p_i$ and $p_j$ give the same correct and incorrect response by chance.[6]

**Confidence intervals.** Unfortunately, the confidence interval of $c_{obs_{i,j}}$ in the scatter-plot of Figure 1a is not trivial to obtain. [28] used a standard binomial confidence interval. This is, however, only a very rough estimate of the true confidence interval since the position on the x-axis ($c_{exp}$) itself is also estimated from the data and thus influenced by variation. We sample data for the null hypothesis of independent observers and calculate the corresponding 95% percentiles (cf. Figure 2). This process is described in Section S.3 in the appendix.

**Bounds.** Confidence intervals allow to investigate hypotheses. In addition, theoretical bounds might help to assess the degree of the observed consistency not being due to chance: a data point close or at the bound has maximum distance to the diagonal for a given value of $c_{exp}$. For this end we have calculated bounds of $c_{obs}$ as an additional diagnostic tool. The influence of these bounds on the confidence intervals is visualised in Figure 2.

Ideally, we also want to express the bounds of $c_{obs}$ directly as a function of $c_{exp}$. The analytical derivation of the bounds below can be found in the Appendix (S.2) and are visualised in Figure 2.

$$0 \leq c_{obs_{i,j}} \leq 1 - \sqrt{1 - 2c_{exp_{i,j}}} \qquad \text{if } c_{exp_{i,j}} \leq 0.5, \tag{2}$$

$$\sqrt{2c_{exp_{i,j}} - 1} \leq c_{obs_{i,j}} \leq 1 \qquad \text{if } c_{exp_{i,j}} \geq 0.5. \tag{3}$$

### 2.2 Error consistency measured by Cohen's kappa

$c_{obs_{i,j}}$ described above quantifies the observed error overlap between observers $i$ and $j$. In order to obtain a single behavioural score for error consistency, that is, one disentangled from accuracy[7], we need to discount for error overlap by chance $c_{exp_{i,j}}$. This is solved by Cohen's $\kappa$ [29] with which we

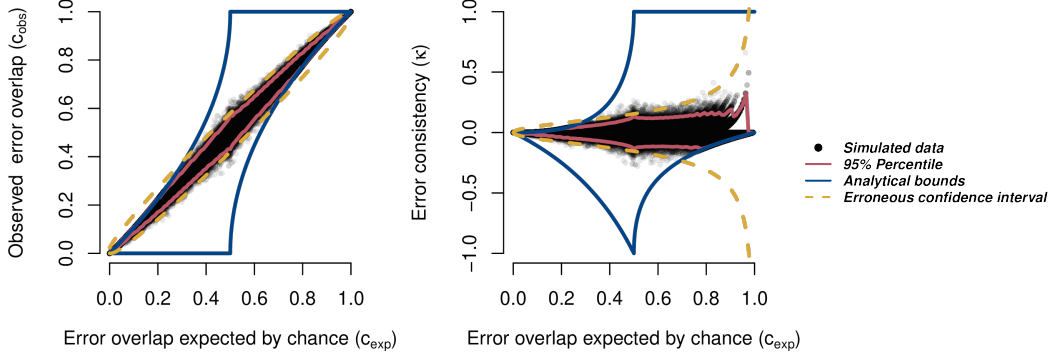

Figure 2: Simulated data of $c_{exp}, c_{obs}$ and $\kappa$ for 160 trials under the assumption of independent decision makers. Analytical bounds and 95% percentile derived from the simulation of 100,000 experiments do not align with the often reported erroneous confidence interval.

measure error consistency:

$$\kappa_{i,j} = \frac{c_{obs_{i,j}} - c_{exp_{i,j}}}{1 - c_{exp_{i,j}}}. \tag{4}$$

We do not include a comparison of $\kappa$ to the (Pearson) correlation coefficient since it has been shown that correlation is not a suitable measure of agreement [32, 33].

**Confidence intervals.** Confidence intervals of the average $\kappa$ of groups, such as the average error consistency of humans vs. humans in Figure 1c, are based on the empirical standard error of the mean and a normal distribution assumption of the average error consistency (a numerical simulation of binomial observers confirmed that this assumption is valid here). Analogous to the observed consistency we use a sampling approach to obtain confidence intervals of $\kappa$ given $c_{exp}$, see S.3 for details. This is necessary since the original confidence approximation interval derived by Cohen [29] (yellow dashes for error consistency in Figure 2) were later shown to be erroneous [34, 35].[8] While a corrected approximate version for individual kappas does exist [34, 38], there is to our knowledge no analytical or approximate confidence interval for $\kappa$ given $c_{exp}$, and hence our sampling approach.

**Bounds.** The following bounds show the limits of $\kappa$ given a specific value of $c_{exp}$, please see Section S.2 for the derivation and Figure 2 for visualisation[9]:

$$\frac{-c_{exp_{i,j}}}{1 - c_{exp_{i,j}}} \leq \kappa_{i,j} \leq \frac{1 - \sqrt{1 - 2c_{exp_{i,j}}} - c_{exp_{i,j}}}{1 - c_{exp_{i,j}}} \qquad \text{if } c_{exp_{i,j}} \leq 0.5, \tag{5}$$

$$\frac{\sqrt{2c_{exp_{i,j}} - 1} - c_{exp_{i,j}}}{1 - c_{exp_{i,j}}} \leq \kappa_{i,j} \leq 1 \qquad \text{if } c_{exp_{i,j}} \geq 0.5. \tag{6}$$

## 2.3 Experimental methods

**Stimuli: motivation.** Exemplary stimuli are visualised in Figure 3. We tested both "vanilla" images (plain unmodified colour images from ImageNet [40]) and three different types of out-of-distribution (o.o.d.) images. The motivation for using o.o.d. images is the following: Significant progress in neuroscience—e.g., discovering receptive fields of simple and complex cells—was made using "unnatural" bar-like stimuli. In deep learning, adversarial examples and texture bias were discovered by testing models on (unnatural) images different than the training data. Hence, we can learn a lot about the inner workings of a system by probing it with appropriate "artificial" stimuli [41, 42]; [4] even argues that o.o.d. testing is a necessity for drawing reliable inferences about a model's strategy. Standard ImageNet images (where human and pre-trained CNN accuracies are both very high and similar, $.960 \pm .036\%$) are included as a baseline condition.

**Stimuli: method details.** [11] tested N=10 human observers in their cue conflict, edge and silhouette experiments. Starting from normal images with a white background, different image manipulations were applied. For *cue conflict* images, the texture of a different image was transferred to this image using neural style transfer [43], creating a texture-shape cue conflict with a total of 1280 trials per observer and network. For *edge* stimuli, a standard edge detector was applied to the original images to obtain line-drawing-like stimuli (160 trials per observer). *Silhouette* stimuli were created by filling the outline of an object with black colour, leaving just the silhouette (160 trials per observer).[10] Lastly, *ImageNet* stimuli were standard coloured ImageNet images; we used the behavioural data (N=2 observers) and stimuli from [44] for this experiment.

**Paradigm.** In order to compare the error consistency of two perceptual systems (e.g. CNNs and humans), those two systems a) need to be evaluated on the exact same stimuli and b) need to be in a regime with neither perfect accuracy nor chance-level performance. We found the publicly available stimuli and data from [11] to be an ideal test case. [11] compared object recognition abilities of humans and algorithms in a carefully designed psychophysical experiment. After a 200 ms presentation of a $224 \times 224$ pixels image, observers had 16 categories to choose from (e.g. `car`, `dog`, `chair`). For ImageNet-trained networks, categorisation responses for 1,000 fine-grained classes were mapped to those 16 classes using the WordNet hierarchy [45]. In order to obtain the probability of a broad category (e.g. `dog`), response probabilities of all corresponding fine-grained categories (e.g. all ImageNet dog breeds) were averaged using the arithmetic mean.[11]

**Convolutional Neural Networks.** Human responses were compared against classification decisions of all available CNN models from the PyTorch model zoo (for `torchvision` version 0.2.2) and against a recurrent model, CORnet-S [46]. All CNNs were trained on ImageNet. Details here: S.5. Additionally, we analysed the relationship between model shape bias (induced by training on Stylized-ImageNet) and error consistency with human observers: S.6.

## 3 Results

If two perceptual systems or decision makers implement the same strategy they can be expected to systematically make errors on the same stimuli. In the following, we show how *error consistency* can be used within visual object recognition to compare algorithms with humans (Section 3.1) and algorithms with algorithms (Section 3.2).

### 3.1 Comparing algorithms with humans: investigating whether better ImageNet models show higher error consistency with human behavioural data

In deep learning, there is a strong linear relationship between ImageNet accuracy and transfer learning performance [47]; in computational neuroscience, better categorisation accuracy improves the prediction of neural firing patterns [48]. But do better performing ImageNet models also make more human-like errors?

**Error consistency vs. model performance.** In Figure 3, we analyse the error consistency between human observers and sixteen standard ImageNet-trained CNNs. We find that humans to humans show a fair degree of consistency w.r.t. individual stimuli. That is, their agreement on which cats or chairs or cars are easy/hard to categorise is well beyond chance. Interestingly, CNN-to-CNN consistency is even higher than human-to-human consistency in all three experiments. This occurs despite the fact that human accuracies are higher than CNN accuracies across experiments: for instance in the silhouette experiment, the average human accuracy is 0.75 whereas the average CNN accuracy is 0.54 (see Table 1, supplementary information). However, the consistency between CNNs and humans is close to zero for two experiments (cue conflict stimuli and line drawings); a linear model fit indicates no improvement with better ImageNet validation accuracy: $F(1, 158) = 0.086, p = .769, R^2 = 0.001$ for cue conflict and $F(1, 158) = 0.478, p = .491, R^2 = 0.003$ for line drawing stimuli. For silhouettes, there is a significant *positive* relationship between ImageNet accuracy and error consistency with $F(1, 158) = 53.530, p = 1.21 \cdot 10^{-11}, R^2 = 0.253$; for ImageNet images, on the other hand, there is a significant *negative* relationship between top-5 accuracy and error consistency with $F(1, 30) = 8.162, p = .008, R^2 = 0.214$.

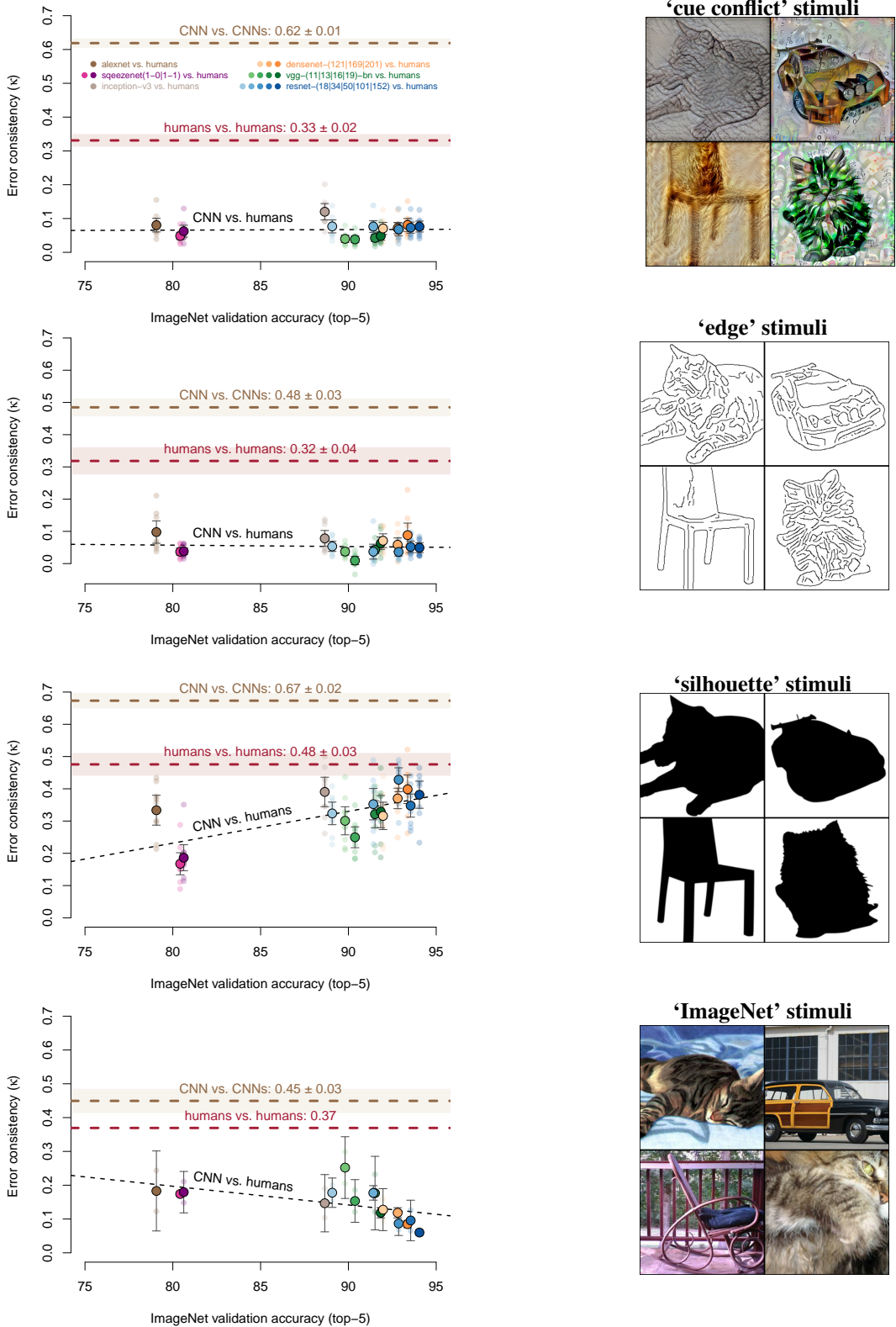

Figure 3: Do better ImageNet models make more human-like errors? Error consistency vs. top-5 ImageNet validation accuracy for four experiments: cue-conflict, edges, silhouettes and standard ImageNet images (exemplary stimuli are visualised on the right). Model colours as in Figure 4a; similar colours indicate same model family. Dashed black lines plot a linear model fit. Whiskers and colored tube show 95% confidence intervals around the mean. Small transparent circles indicate error consistency between a CNN and an individual human observer; mean consistency is shown as a larger saturated circle.

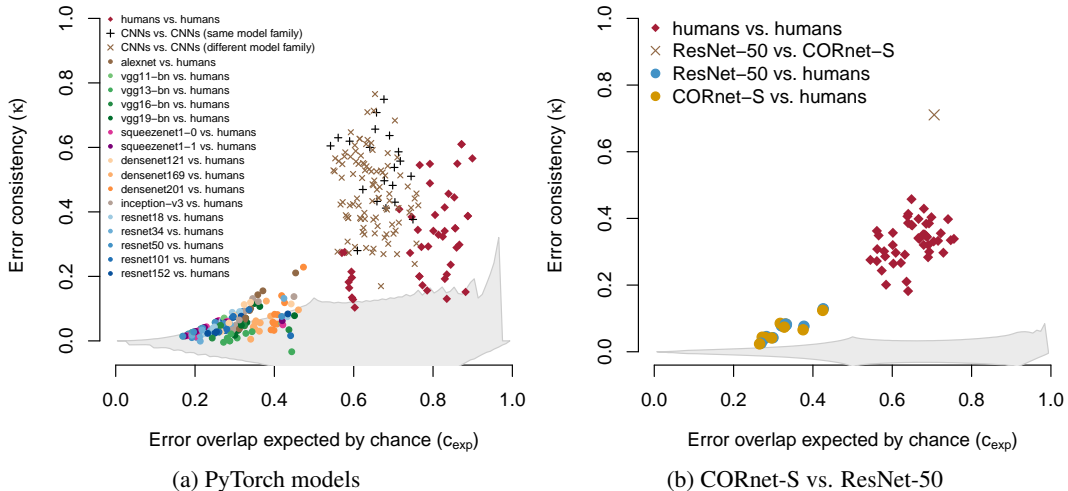

(a) PyTorch models      (b) CORnet-S vs. ResNet-50

Figure 4: **(a)** How is error consistency influenced by model architecture? PyTorch models tested on edge stimuli (160 trials per observer). **(b)** Recurrent CORnet-S behaves just like a standard feedforward ResNet-50 on cue conflict stimuli (1280 trials). Shaded areas indicate a simulated 95% percentile for consistency by chance.

We conclude that there is a substantial algorithmic difference between human observers and the investigated sixteen CNNs: humans and CNNs are very likely implementing different strategies. This difference is narrowing down for silhouette stimuli, whereas it is as big as ever for cue conflict, line drawing and ImageNet stimuli: AlexNet from 2012 is just as error-consistent as recent models. Our results are in stark contrast to the observation that better ImageNet models appear to be better models of the primate visual cortex, even if they better predict neural activity [48].

**Error consistency vs. model architecture.** We were surprised to see that the consistency between different CNNs is even higher than the consistency between different human observers. In Figure 4a, we investigate the degree to which this CNN-CNN consistency is influenced by similarities in model architecture. When distinguishing between models from the same architecture family (e.g., all ResNet models) and models from a different model family (e.g., ResNet vs. VGG) we observe that even though models from the same family score higher on average, model-to-model consistency is generally very high.[12] In line with these results, [27] also reported extremely high similarity between different models on the ImageNet test set. This might shed some light on the finding that many trained and fitted CNNs predict neural data similarly well, largely irrespective of architecture [49]. Interestingly, the highest observed error consistency ($\kappa = 0.793$) occurs for DenseNet-121 vs. ResNet-18: two models from a different model family with different depth (121 vs. 18 layers) and different connectivity. High error consistency between different CNNs suggests that using CNNs as an ensemble may currently be less effective than desirable, since ensembles benefit from independent (rather than consistent) models. It remains an open question why even multiple instances of a single model (trained with a different random seed) *internally* often differ substantially [50, 51], yet in spite of large architectural differences across models and model families, all CNNs that we investigated seem to be implementing fairly similar strategies.

### 3.2 Comparing algorithms with algorithms: the "current best model of the primate ventral visual stream" behaves like a vanilla ResNet-50 according to error consistency analysis

In order to understand how object recognition is achieved in brains, a necessary—but not sufficient—pre-requisite are quantitative metrics to track improvements and models that improve on those metrics. [46] went an important step in both directions by proposing `Brain-Score`, a benchmark where models can be ranked according to a number of metrics, for instance how well their activations predict how biological neurons fire when primates see the same images as an ImageNet-trained CNN. Using this benchmark, the authors tested hundreds of architectures to develop CORnet-S, a brain-inspired recurrent neural network. CORnet-S is able to capture recurrent dynamics (so-called object solution times) of monkey behaviour and achieves previously unmatched performance on

`Brain-Score` while retaining good ImageNet performance (73.1% top-1). These results, in the author's words, "establish CORnet-S, a compact, recurrent ANN, as the current best model of the primate ventral visual stream" performing "brain-like object recognition" [46, p. 1]. Building such a model is an exciting undertaking and, as perhaps indicated by the highly competitive selection as an "Oral" contribution to NeurIPS 2019, an endeavour that sparked considerable excitement at the intersection of the neuroscience and machine learning communities. But how much is behavioural consistency improved in comparison to a baseline model (ResNet-50)? This is exactly the type of question that can be answered with the help of our error consistency analysis.

Figure 4 shows that CORnet-S shares only slightly above-chance error consistency with most human observers—even the highest CORnet-S-to-human error consistency is lower than the lowest human-to-human error consistency. However, there is no improvement whatsoever over a ResNet-50 baseline: Cohen's $\kappa$ for CNN-human consistency is very low for both models (.068: ResNet-50; .066: CORnet-S) compared to .331 for human-human consistency. Perhaps worse still, AlexNet from 2012 has higher error consistency than CORnet-S (.080). CNN-CNN consistency between CORnet-S and ResNet-50 is exceptionally high (.711), many datapoints even overlap exactly—a pattern confirmed by additional experiments in the appendix (Figures SF.5, SF.6 and SF.7), where we also perform a more detailed comparison to all six `Brain-Score` metrics (Figures SF.9, SF.10, SF.11 and SF.12 showing, if at all, only a weak relationship between error consistency and `Brain-Score` metrics). This indicates that CORnet-S is likely implementing a very different strategy than the human brain: in our analysis, CORnet-S has more behavioural similarities with a standard feedforward ResNet-50 than with human object recognition.[13] This provides evidence that recurrent computations—often argued to be one of the key missing ingredients in standard CNNs towards a better account of biological vision [46, 52–56]—do not necessarily lead to different behaviour compared to a purely feedforward CNN. It is still an open question to determine the conditions under which recurrence provides advantages over feedforward networks. Recent evidence seems to indicate that recurrence may be especially useful for difficult images [57–59].

Overall, the observed discrepancy between the leading score of CORnet-S on `Brain-Score` and its similarity to a standard ResNet-50 according to error consistency analysis points to the decisive importance of metrics: CORnet-S was mainly built for neural predictivity and while it scores very well on a number of other benchmarks, such as capturing object solution times and even a previously reported behavioural error analysis [26], it performs poorly on the behavioural metric reported here, *trial-by-trial error consistency*. New metrics to scrutinise models will hopefully lead to an improved generation of models, which in turn might inspire ever-more challenging analyses. An ideal model of biological object recognition would score well on multiple metrics (both neural and behavioural data, an important idea behind `Brain-Score`), including on metrics that the model was not directly optimised for.

## 4 Conclusion

Error consistency is a quantitative analysis for comparing strategies/methods of black-box decision makers—be they brains or algorithms. Accuracy alone is insufficient for distinguishing between strategies: two decision makers may achieve similar accuracy with very different strategies. In contrast to aggregated metrics (averaging across trials/stimuli and observers/networks), error consistency measures behavioural errors on a fine-grained level following the idea of "molecular psychophysics" [18]. Using error consistency we find:

- Irrespective of architecture, CNNs are remarkably consistent with one another

- The consistency between humans and CNNs, however, is little beyond what can be expected by chance alone, indicating that CNNs still employ very different perceptual mechanisms and "brain-like machine learning" may be still but a distant dream (cf. [60])

- Recurrent CORnet-S, termed the "current best model of the primate ventral visual stream", fails to capture essential characteristics of human behavioural data and instead behaves effectively like a standard feedforward ResNet-50 in our analysis.

Taken together, error consistency analysis suggests that the strategies used by human and machine vision are still very different—but we envision that error consistency will be a useful analysis in the quest to understand complex systems, be they CNNs or the human mind and brain.

## Broader Impact

*Error consistency* is a statistical analysis for measuring whether two or more decision makers make similar errors. Like any statistical analysis, it can be used for better or worse. For instance, as a very simple example, calculating the *mean* of a number of observations can be used to quantify a world-wide temperature increase caused by human carbon emissions [61, 62] (positive impact). However, calculating the mean could just as well be utilised by authoritarian governments to obtain an aggregated credit score of "social"—i.e., conformist—behaviour (negative impact) [63]. Concerning error consistency, we could envisage the following broader impact.

**Potential positive impact.** Quantifying differences between decision making strategies can contribute to a better understanding of algorithmic decisions. This improves model interpretability, which is a scientific goal by itself but also closely linked to societal requirements like accountability of algorithmic decision making and the "right to explanation" in the European Union [64]. Furthermore, calculating the error consistency between humans and CNNs can be used for fact-checking overly hyped "human-like AI" statements, e.g. by startups. We argue that human-level accuracy does not imply human-like decision making, which might contribute to increased rigour in model evaluation.

**Potential negative impact.** While not intended to cause any harm, quantifying differences between individuals can be used to identify group-conform and outlier behaviour. Furthermore, measuring error consistency between machines and humans might be used to quantify progress towards building machines that mimic human decision making on certain tasks. While this might sound exciting to a scientist, it very likely sounds a lot more frightening from the perspective of someone losing their job because a machine would then be capable of doing the same work more cheaply. Depending on the complexity of the task, this may not be a problem in the near future but, given current trends in the use of machine learning for automation, perhaps in the distant future.

### Acknowledgments & funding disclosure

Funding was provided, in part, by the Deutsche Forschungsgemeinschaft (DFG, German Research Foundation) – project number 276693517 – SFB 1233, TP 4 Causal inference strategies in human vision (K.M. and F.A.W.). The authors thank the International Max Planck Research School for Intelligent Systems (IMPRS-IS) for supporting R.G; and the German Research Foundation through the Cluster of Excellence "Machine Learning—New Perspectives for Science", EXC 2064/1, project number 390727645, for supporting F.A.W. The authors declare no competing interests.

We would like to thank Silke Gramer and Leila Masri for administrative and Uli Wannek for technical support; and David-Elias Künstle, Bernhard Lang, Maximus Mutschler as well as Uli Wannek for helpful comments. We thank Kubilius et al. [46] for making their implementation of CORnet-S publicly available. Furthermore, we would like to thank Jonas Kubilius and Martin Schrimpf for feedback and many valuable suggestions.

### Author contributions

Based on ideas from [65] and [28], R.G. first applied trial-by-trial analysis ideas to CNNs. Thereafter, all three authors jointly initiated the project. R.G. and K.M. jointly led the project. K.M. derived the bounds, performed and visualised the simulations and acquired the Brain-Score data (with input from R.G.). The CNN experiments were performed, analysed and visualised by R.G. (with input from K.M.). F.A.W. provided guidance, feedback, and pointed out the link to molecular psychophysics. All three authors planned and structured the manuscript. R.G. and K.M. wrote the paper with active input from F.A.W.

## Footnotes

[1]Blog post summary: https://medium.com/@robertgeirhos/are-all-cnns-created-equal-d1 3a33b0caf7

[2]Note again the parallel in neuroscience, even for very simple brains: The nervous system of the nematode *C.elegans* is basically known in its entirety— still it is not fully understood how the (comparatively) complex behaviour of *C.elegans* is brought about by the biological "hardware" [5].

[3]For a discussion of this terminology we refer to Section S.1 in the appendix

[4][17] found surprising similarities for self-supervised vs. supervised CNNs using error consistency.

[5]For human observers the order of presentation can make a (typically small) difference as human observers exhibit serial dependencies and other non-stationarities [18, 30]. Participants, e.g., may make more errors or lapses towards the end of an experiment due to fatigue [31]), and it is thus recommended to randomly shuffle presentation order for each participant to avoid such a "trivial" consistency of errors. Luckily, non-stationarities are usually only problematic if the signal levels are low, i.e. near chance performance.

[6]Note that $c_{exp} > 0.5 \iff p_1, p_2 > 0.5 \lor p_1, p_2 < 0.5$, see also Figure SF.2 in the appendix.

[7]In fact, error consistency is an accuracy corrected metric, see S.4 in appendix

[8]This erroneous confidence interval is still used in many publications, including very influential ones [36, 37].

[9]Bounds of kappa depending on $c_{obs}$ instead of $c_{exp}$ can be found in [39].

[10]For parametrically distorted images (Appendix, Figure SF.7) we used the stimuli from [44].

[11]This aggregation is optimal. A derivation is included in the appendix of the arXiv version `v3` of [44].

[12]Results for two other experiments are plotted in the appendix, Figure SF.3.

[13]Interestingly, CORnet-S and ResNet-50 also score fairly similarly on a few metrics of `Brain-Score`.

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
