[Supplementary Material]

# Supplementary Material

Code and data to reproduce results and figures are available from `https://github.com/wichmann-lab/error-consistency`.

The supplementary material is structured as follows. We start with terminology in Section S.1, afterwards we derive bounds of $c_{obs}$ and kappa in Section S.2 (limiting possible consistency), followed by a description of how we simulated the confidence intervals for $c_{exp}$ and kappa under the null hypothesis of independent observers in Section S.3. Finally, we provide method details for `Brain-Score` and the evaluated CNNs in Section S.5 and report accuracies across experiments in Table 1.

In addition to method details, we provide extended experimental results in Figure SF.3 (error consistency of all PyTorch models for cue conflict and edge stimuli) as well as Figures SF.5, SF.6, SF.7, SF.8 (detailed analyses of CORnet-S vs. ResNet-50). Figures SF.9, SF.10 and SF.11 and SF.12 (investigating the relationship between `Brain-Score` metrics and error consistency).

Furthermore, Figure SF.4 visualises qualitative error differences by plotting which stimuli were particularly easy for humans and CNNs, respectively.

## S.1 Terminology: "error consistency"

We would like to briefly clarify the name *error consistency*. Our analysis helps to compare the consistency of two decision makers. Two decision makers necessarily show some degree of consistency due to chance agreement. Error consistency helps to examine whether the two decision makers show significantly more consistency than expected by chance by analysing behavioural error patterns. However, this analysis takes into account not only the consistency of errors but also the consistency of correctly answered trials, hence 'error consistency' may sound imprecise at first. Nonetheless, we believe that the term captures the most crucial aspect of this analysis: Humans and CNNs —which are particularly well suited for our analysis—are often close to ceiling performance or at least have high accuracies. Thus trials where the decision makers agree do not provide much evidence for distinguishing between processing strategies. In contrast, the (few) errors of the decision makers are the most informative trials in this respect: Hence the name error consistency.

## S.2 Derivation of bounds for $c_{obs}$ and kappa given $c_{exp}$

How much observed consistency can we expect at most for a given expected consistency? We assume two independent observers $i$ and $j$ with accuracies $p_i$ and $p_j$. For given $p_i, p_j$ only a certain range of $c_{obs}$ is possible:

$$c_{obs_{max}} = 1 - |p_i - p_j| \text{ and } c_{obs_{min}} = |p_j + p_i - 1|. \tag{7}$$

Ideally, we also want to express the bounds of $c_{obs}$ directly as a function of $c_{exp}$. We obtain the following bounds:

$$0 \leq c_{obs_{i,j}} \leq 1 - \sqrt{1 - 2c_{exp_{i,j}}} \qquad if \ c_{exp_{i,j}} < 0.5, \tag{8}$$

$$\sqrt{2c_{exp_{i,j}} - 1} \leq c_{obs_{i,j}} \leq 1 \qquad if \ c_{exp_{i,j}} \geq 0.5. \tag{9}$$

These bounds are visualised in Figure 2.

The derivation is as follows. We distinguish between two cases.

**Case 1:** $p_i \leq 0.5 \ \& \ p_j \leq 0.5$ or $p_i \geq 0.5 \ \& \ p_j \geq 0.5 \iff c_{exp_{i,j}} \geq 0.5$

The expected consistency then lies in the interval of $[0.5, 1]$, see Figure SF.2. First we calculate the upper bound $b_{obs_{max}}$ given $c_{exp_{i,j}}$. Please note that a specific $c_{exp_{i,j}}$ can be obtained by multiple combinations of values for $p_i$ and $p_j$. For a given $c_{exp_{i,j}}$ we choose $p_j = p_i$. We can calculate the exact value of $p_i$ in this case with eq. (1). However since $p_j = p_i$ we get with eq. (7) that $b_{obs_{max}} = 1$. Thus we directly obtain from eq. (7) that the upper bound of $c_{obs_{i,j}}$ is always 1 for all $c_{exp_{i,j}}$ in the interval [0.5, 1].

It is a bit more challenging to derive the lower bound $b_{obs_{min}}$ given $c_{exp_{i,j}}$. Using equation (7) and (1) we obtain

$$b_{obs_{min}} = p_i + \frac{c_{exp_{i,j}} + p_i - 1}{2p_i - 1} - 1. \tag{10}$$

Setting $\frac{\partial b_{obs_{min}}}{\partial p_i} = 0$ to find the minimum results in

$$p_{i_{min}} = \frac{1}{2} \pm \sqrt{\frac{1}{4} - \frac{-2c_{exp_{i,j}} + 2}{4}}. \tag{11}$$

We only take the positive term in eq. (11) since $p_i > 0.5$ by definition. Checking the second order derivative confirms a minimum. Finally using equation eq. (11) with eq. (10) we calculate

$$b_{obs_{min}} = \sqrt{2c_{exp_{i,j}} - 1}, \text{ thus} \qquad (12)$$

$$\sqrt{2c_{exp_{i,j}} - 1} \leq c_{obs_{i,j}} \leq 1. \qquad (13)$$

**Case 2:** $p_i > 0.5$ & $p_j < 0.5$ or $p_i < 0.5$ & $p_j > 0.5 \Longleftrightarrow c_{exp_{i,j}} < 0.5$

The expected consistency then lies in the interval of $[0, 0.5[$, see Figure SF.2. This case is point symmetric to the right part. Thus we obtain for the bounds of the left part

$$b_{obs_{max2}} = 1 - b_{obs_{min}}(1 - c_{exp_{i,j}}), \qquad (14)$$

$$b_{obs_{min2}} = 0 \text{ and finally} \qquad (15)$$

$$0 \leq c_{obs_{i,j}} \leq 1 - \sqrt{1 - 2c_{exp_{i,j}}}. \qquad (16)$$

**Bounds for kappa** If we plug in the bounds of $c_{obs_{i,j}}$ into the equation of kappa, we obtain the following bounds for kappa:

$$\frac{-c_{exp_{i,j}}}{1 - c_{exp_{i,j}}} \leq \kappa_{i,j} \leq \frac{1 - \sqrt{1 - 2c_{exp_{i,j}}} - c_{exp_{i,j}}}{1 - c_{exp_{i,j}}} \qquad \text{if } c_{exp_{i,j}} < 0.5, \qquad (17)$$

$$\frac{\sqrt{2c_{exp_{i,j}} - 1} - c_{exp_{i,j}}}{1 - c_{exp_{i,j}}} \leq \kappa_{i,j} \leq 1 \qquad \text{if } c_{exp_{i,j}} \geq 0.5. \qquad (18)$$

## S.3 Calculating 95% percentiles of observed overlap and kappa for the null hypothesis of independent observers given an expected consistency

Here we describe the procedure to calculate 95% percentiles of $\kappa$ and $c_{obs}$.

Our null hypothesis is that two decision makers are independent. Assuming independence, we can easily simulate these two observers. Based on $p_i, p_j$ (the accuracies of decision makers $i$ and $j$) we sample $n$ trials and calculate $c_{exp_{i,j}}, c_{obs_{i,j}},$ and $\kappa_{i,j}$ accordingly based on these simulated values. This process is repeated systematically for different $p_i$ and $p_j$. For this purpose we sample a grid of 4200 x 4200 points in the range $[[0, 1], [0, 1]]$. For each individual combination of $p_i$ and $p_j$, the sampling is repeated five times, thus in total we simulate $4200 \times 4200 \times 5 = 88,200,000$ values.[14]

The grid is not divided equally. 66% of $p_i$ and $p_j$ are located in the upper and lower 15% of the domain. This is important because kappa diverges for large values of $c_{exp}$ (small and large values of $p_i$ and $p_j$); thus a dense sampling is necessary there.

Based on these simulated data we obtain 95% percentiles for $c_{obs}$ and $\kappa$. We binned the data in 1% steps and used the standard quantile-function of R (type 7, see [66]). It is important to note that we have only a small number of trials (160 or 1280).[15] Therefore $c_{obs}$ can take a maximum number of 161 or 1281 values respectively. The range of uniquely observed values is very small for a given $c_{exp}$. This implies that the accuracy of our percentiles is limited for data points that are very close to the quantiles. However, this does not influence our findings.

Please note that the denominator of kappa gets very small for high values of $c_{exp}$. Thus we see some instability of kappa towards high expected consistencies. Figure SF.1 shows diagnostic plots for both cases.

## S.4 Disentangling of Error consistency and Accuracy

Our argument for the disentanglement between kappa and accuracy is as follows. For independent observer no correlation between accuracy and kappa is observed, e.g. In Figure 2b, $\kappa$ and $c_{exp}$ [16] are not correlated (r=-0.00015, $p > 0.05$). As expressed by the bounds in Figure 2, $\kappa$ is limited by accuracy. If two observers have an accuracy for 90%, only certain levels of (dis-)agreement are possible. Error consistency (measured by $\kappa$) aims to correct for accuracy and thus in our experiments different kinds of correlations between error consistency and overall accuracy occur. We observe zero correlation in (Figures 3a, 3b) and positive correlation in Figure 3c. In Figure 3d we observe a negative correlation between accuracy and error consistency. We conclude that there is

Figure SF.1: Simulated data of $c_{exp}, c_{obs}$ and $\kappa$ for 160 (top) and 1280 (bottom) trials per block. Black dots show 100.000 randomly drawn blocks from our simulation. Blue lines show analytical bounds. Red lines show the 95% percentiles. Orange dashed lines show the wrong binomial confidence interval (left) and the erroneous confidence interval for $\kappa$ (right) reported in many papers.

*no* correlation between consistency ($\kappa$) and accuracy for independent observers whilst for dependent (consistent) observers correlations are possible. Kappa corrects for accuracy but is not independent from it.

## S.5   Method details for Brain-Score and CNNs

Human responses were compared against classification decisions of all available CNN models from the PyTorch model zoo (for `torchvision` version 0.2.2) [67], namely `alexnet`, `vgg11-bn`, `vgg13-bn`, `vgg16-bn`, `vgg19-bn`, `squeezenet1-0`, `squeezenet1-1`, `densenet121`, `densenet169`, `densenet201`, `inception-v3`, `resnet18`, `resnet34`, `resnet50`, `resnet101`, `resnet152`.   For the VGG model family [68], we used the implementation with batch norm. CORnet-S, an additional recurrent model [46] analysed in Section 3.2, was obtained from the author's github implementation.[17]   The comparison to `Brain-Score` in Figures SF.9, SF.10, SF.11 and SF.12 uses `Brain-Score` values obtained from the Brain-Score website(date of download: April 17, 2020) and error consistency values obtained by us. Note that the model implementations differ slightly: we consistently used PyTorch models whereas `Brain-Score` tested models from a few different frameworks (the full list can be seen here). Namely, `squeezenet1-0`, `squeezenet1-1`, `resnet18`, `resnet-34` are identical (PyTorch); the VGG models use Keras instead (without batch norm) and so do the `Brain-Score` DenseNet models; `inception_v3`, `resnet50_v1`, `resnet101_v1`, `resnet152_v1` are TFSlim models. Since model implementations usually differ slightly

across frameworks, a small variation in the results can be expected depending on the chosen model and framework.

## S.6 Error consistency of shape-biased models

We analyzed three CNNs with different degrees of stylized training data from [11]. Model shape bias predicts human-CNN error consistency for cue conflict stimuli, indicating that networks basing their decisions on object shape (rather than texture) make more human-like errors:

| model shape bias (%) | 20.5 | 21.4 | 34.7 | 81.4 |
|---|---|---|---|---|
| human-CNN consistency ($\kappa$) | .066 | .068 | .098 | .195 |

| | observer / model | cue conflict | edge | silhouette |
|---|---|---|---|---|
| 1 | subject-01 | 0.69 | 0.89 | 0.80 |
| 2 | subject-02 | 0.76 | 0.94 | 0.66 |
| 3 | subject-03 | 0.84 | 0.93 | 0.80 |
| 4 | subject-04 | 0.62 | 0.84 | 0.78 |
| 5 | subject-05 | 0.85 | 0.89 | 0.77 |
| 6 | subject-06 | 0.82 | 0.93 | 0.72 |
| 7 | subject-07 | 0.76 | 0.81 | 0.76 |
| 8 | subject-08 | 0.78 | 0.96 | 0.64 |
| 9 | subject-09 | 0.86 | 0.61 | 0.76 |
| 10 | subject-10 | 0.77 | 0.92 | 0.85 |
| 11 | alexnet | 0.19 | 0.29 | 0.43 |
| 12 | vgg11-bn | 0.12 | 0.14 | 0.46 |
| 13 | vgg13-bn | 0.12 | 0.25 | 0.36 |
| 14 | vgg16-bn | 0.14 | 0.22 | 0.47 |
| 15 | vgg19-bn | 0.15 | 0.28 | 0.46 |
| 16 | squeezenet1-0 | 0.14 | 0.15 | 0.24 |
| 17 | squeezenet1-1 | 0.17 | 0.14 | 0.29 |
| 18 | densenet121 | 0.19 | 0.24 | 0.42 |
| 19 | densenet169 | 0.21 | 0.33 | 0.53 |
| 20 | densenet201 | 0.21 | 0.38 | 0.51 |
| 21 | inception-v3 | 0.27 | 0.28 | 0.54 |
| 22 | resnet18 | 0.19 | 0.20 | 0.47 |
| 23 | resnet34 | 0.19 | 0.16 | 0.45 |
| 24 | resnet50 | 0.18 | 0.14 | 0.54 |
| 25 | resnet101 | 0.20 | 0.24 | 0.49 |
| 26 | resnet152 | 0.21 | 0.21 | 0.56 |
| 27 | cornet-s | 0.18 | 0.25 | 0.46 |

Table 1: Accuracies for human observers and CNNs for all three experiments. In the cue conflict experiment case, an answer is counted as correct in this table if this answer corresponds to the correct shape category (other choices are possible).

Figure SF.2: Values that $c_{exp}$ can take depending on $p_i$ and $p_j$ for two independent observers.

(a) Cue conflict stimuli

(b) Edge stimuli

Figure SF.3: Error consistencs vs. expected error overlap for all PyTorch models.

(a) Cue conflict stimuli

(b) Edge stimuli

(c) Silhouette stimuli

Figure SF.4: "Easy" stimuli for humans and CNNs. For each experiment, the images in the top row were those that most humans correctly classified. In the bottom row: stimuli that most CNNs correctly classified. If there were more than five images where humans were very accurate on, we here selected those where CNNs were the least accurate, and vice versa. ImageNet stimuli are not visualised due to image permission reasons.

Figure SF.5: Shape bias of CORnet-S and ResNet-50 in comparison to human observers. Human observers categorise objects by shape rather than texture [11], which differentiates them from standard ImageNet-trained CNNs like ResNet-50 (categorising predominantly by texture). In this experiment, CORnet-S again behaves similarly to ResNet-50 but does not show a human-like shape bias as would be expected for an accurate model of human object recognition. Small bar plots on the right indicate accuracy (answer corresponds to either correct texture category or correct shape category). This pattern was also observed by Hermann and Kornblith [69], who performed a detailed investigation of the factors that influence model shape bias.

(a) Edge stimuli

(b) Silhouette stimuli

Figure SF.6: Error consistency of CORnet-S vs. ResNet-50 for edge and silhouette stimuli.

Figure SF.7: Classification accuracy on parametrically distorted images for ResNet-50, CORnet-S and human observers. Again, CORnet-S behaves like a ResNet-50 rather than like human observers.

Human average        ResNet-50        CORnet-S

Figure SF.8: Confusion matrices for humans, ResNet-50 and CORnet-S. Different rows correspond to different experiments. Top row: cue conflict stimuli, second row: edge stimuli, third row: silhouette stimuli, last row: ImageNet stimuli.

Figure SF.9: Error consistency vs. `Brain-Score` metrics for PyTorch models, "cue conflict" stimuli.

Figure SF.10: Error consistency vs. `Brain-Score` metrics for PyTorch models, "edge" stimuli.

Figure SF.11: Error consistency vs. `Brain-Score` metrics for PyTorch models, "silhouette" stimuli.

Figure SF.12: Error consistency vs. `Brain-Score` metrics for PyTorch models, "ImageNet" stimuli.

## Footnotes

[14]The more values are simulated, the better: we chose the maximum number of samples feasible to simulate on our hardware within reasonable time.

[15]Percentiles for a different number of trials can also be computed with the code that we provide.

[16]Accuracy and $c_{exp}$ are linked as one can see in figure SF.2

[17]`https://github.com/dicarlolab/CORnet`