[Reviews · NeurIPS 2020]

Review 1

Summary and Contributions: This paper first establishes a method for measuring whether two systems make similar errors. While the main analysis is not novel, they develop confidence intervals and analytical bounds to increase the rigour of the analysis. They then apply this method to compare CNNs and humans on visual classification tasks. They find that CNNs are highly consistent with each other, humans are less consistent with other humans, and CNNs and humans have very low consistency with each other, even when a CNN designed as a model of the primate visual system is used. The authors conclude that humans and CNNs use very different strategies for visual classification tasks.

Strengths: This is one of those papers where after reading you may think, oh that was an obvious thing to do and trivial. However, you only think that because this paper is very well written and explained, based on a good idea, and has simple but important findings! The idea of testing strategies based on error consistency is by no means a new one, but to my knowledge has not been done rigorously in the context of CNNs vs humans, which is extremely relevant to the NeurIPS community.

Weaknesses: The key argument that small error consistency implies different strategies was assumed as true throughout this work. The authors provided little evidence for this statement - I would have liked to see more explicit references to and discussion of prior work. Additionally, the authors did not discuss any potential caveats or what "strategy" could refer to. What if humans and CNNs actually use very similar high-level strategies but something like initial color representation results in different inputs being problematic? Additionally, only CNNs trained on ImageNet were used, even though the images used for the error consistency analysis were quite different. CNNs would obviously develop strategies for the specific task/images on which they are trained, whereas humans presumably learn different strategies they can flexibly switch between based on context. I would have liked to see an analysis where at least a few CNNs are trained on these type of images (cue conflict, edge, silhouette) to see if these CNNs are any more consistent with humans. If CNNs would do too well (this seems unlikely), then another task with harder images would be necessary. Finally, the authors briefly discuss why their analysis may produce such different results to Brain-Score in terms of assessing CNN similarity to primate/human visual processing. They noted two reasons that Brain-Score could be problematic (result of testing hundreds of models aka overfitting to neural predictivity and skewed results from penalizing lack of recurrence too strongly). Ideally, the authors would follow up on this and "fix" these problems - just directly compare their CNNs on error consistency vs neural predictivity. Although I realize that would require access to neural data and may be less feasible, I think this would be an important analysis to contextualize their findings. --- UPDATE AFTER AUTHOR REBUTTAL --- The authors addressed many of my concerns and promised to include relevant further analyses in the paper so I am updating my score to an 8.

Correctness: The claims and methods seem correct.

Clarity: This paper is extremely well written. There were almost no points of confusion and I felt that I truly understood the questions, methods, and findings after just one readthrough of the paper.

Relation to Prior Work: Yes, the authors clearly discuss differences from previous contributions.

Reproducibility: Yes

Additional Feedback:


Review 2

Summary and Contributions: This paper addresses the question whether two decision makers, such as humans and neural networks, use the same strategy to solve a given task (e.g. image classification). To that end, the paper introduces the “error consistency” metric, which is known as Cohen’s kappa in psychology. Error consistency measures how similar the errors of two decision makes are, while accounting for the agreement expected by chance. The paper contributes confidence intervals for this statistic. The paper further contributes an empirical study of human and CNN error consistency, concluding that there is high consistency within CNNs and within humans, but not across CNNs and humans, which means that humans and CNNs use different strategies for image classification.

Strengths: The main strength of the paper is that it provides a principled evaluation of CNN performance, including a comparison to human performance, that goes beyond the standard benchmarks. The conclusion that most CNNs use very similar strategies is significant and relevant to the NeurIPS community. This is consistent with previous/concurrent results. It will hopefully encourage the field to explore a wider space of architectures and training schemes. Further, the paper is statistically rigorous in discussing the error consistency metric. Overall, I think this could be a highly impactful paper, if the concerns outlined below can be addressed.

Weaknesses: While I think that the overall idea of the paper is sound and relevant, I have several questions and concerns about the specific approach. Since these concerns may affect some conclusions of the paper, they should be addressed before I can recommend publication. Concern 1: Does kappa really disentangle accuracy and strategy? The paper claims that kappa disentangles error consistency from accuracy. However, the plots of kappa against c_exp suggest that both the magnitude and the variability of kappa are highly correlated with c_exp (which is in turn determined by the relative accuracy of the two decision makers). This heteroscedasticity raises the question if and how kappa can be compared for points that have different values of c_exp. Perhaps, the error consistency metric should be further normalized by the size of the confidence interval at a given value of c_exp. In other words, the final metric should be proportional to the likelihood that datapoint is due to random chance, so that its magnitude can be compared across different values of c_exp. Concern 2: Is it reasonable to compare humans and CNNs? The comparisons between humans and CNNs is confounded by the fact that they have vastly different accuracies on the datasets used for comparison (Supplementary Table 1). In fact, the stimuli from [1] were *designed* to be easy for humans and hard for CNNs (in particular, the shape/texture conflict stimuli). The striking difference in kappa between human-vs-CNN and human-vs-human is only present for the edge and cue-conflict stimuli, where the accuracy difference between humans and CNNs is largest. For Silhouette stimuli, where the accuracy difference is smaller, the difference in kappa is also much smaller (Figure SF.5b). This issue is compounded by the heteroscedasticity (see Concern 1). Two measures are necessary to address this issue: (1) Repeat the experiments on datasets where humans and CNNs have no statistically significant difference in performance, or at least a smaller difference than with the current datasets. (2) Test the statistical significance of the difference in kappa between human-vs-CNN and human-vs-human, taking the heteroscedasticity of the data into account. A further, more general issue with the comparison between CNNs and humans is that CNNs and humans were likely trained on different tasks. Due to the way in which ImageNet labels were collected, CNNs are trained to predict the probability that an image was returned by a particular search query X on Flickr, and verified by a human rater who was asked “Is X present in the image?”. This is probably not how the humans used for the present paper approached the task. If the two decision makers are performing different tasks, this would explain why they appear to use different strategies. This issue should be discussed in the paper. Ideally, the comparisons would also include models trained on different task and data, e.g. the models trained on StylizedImageNet from [1], or self-supervised models (with linear heads trained on ImageNet). Concern 3: Comparison to BrainScore/CORnet. While I sympathize with the authors’ apparent dislike for sensationalist statements about similarity to the brain, I think the criticism of BrainScore and CORnet is overly adversarial and not completely warranted. As far as I understand, BrainScore primarily measures how well the intermediate representations of a CNN can explain neural activity in various brain regions. Only a small part of the score is concerned with the behavioral output, i.e. error consistency. Therefore, the two metrics likely measure orthogonal properties of neural networks. In fact, as cited in [1], some studies show that CNNs match human ventral stream activity mainly due to texture representations, but not shape representations. Given that CNNs are highly texture-biased [1], it is likely that BrainScore measures mainly low-level similarities in representations, while error consistency measures only behavioral similarity at the most abstract level (output). It is therefore plausible that a CNN may have a high BrainScore but low error consistency with humans. It is also plausible to call such a model “the current best model of the primate ventral visual stream”, since “current best” doesn’t mean “perfect”. I would therefore reduce the extensive critical discussion of CORnet and BrainScore and focus on discussing the relationship between representational and behavioral similarity more generally. [1] Geirhos 2018, https://arxiv.org/abs/1811.12231 Edit after rebuttal: The rebuttal addresses my concerns and I updated my score accordingly.

Correctness: I did not find any technical errors. Conceptual issues were addressed above.

Clarity: Yes.

Relation to Prior Work: The paper should discuss the concurrent work https://arxiv.org/pdf/1905.10854.pdf (ICML2020), which comes to the very similar conclusion that all CNNs make similar errors. Otherwise, the discussion of related work is clear and complete.

Reproducibility: Yes

Additional Feedback:


Review 3

Summary and Contributions: In this manuscript, the authors introduce a metric - “error consistency”, to test whether two algorithms make consistent decisions given the same dataset. The authors apply the metric on human and CNN-based object recognition tasks, and found the consistency is low among CNNs and humans, even for the state-of-the-art brain-like model. The authors suggest that CNNs and humans have different strategies for object recognition tasks.

Strengths: To my best knowledge, the application to use error consistency on computation model - human comparison on neurophysiological or behavioral data is novel. The findings that CNNs and humans have high intra but low inter scores is also very intriguing. The manuscript does provide a novel and systematic perspective on how to measure the “brain-likeliness” of the proposed neurocomputational model, which is of interest to broad audience in the community. Experiments and methodologies are solid as well.

Weaknesses: My main concern about the manuscript is the lack of analysis. For study like this, not only the experiment results are important, but understanding underlying reasons/patterns/conditions for the result is more important for me. For example: * Since CNNs vs. CNNs achieve higher scores than humans vs. humans, what are the differences in terms of the errors? * BrainScore has many conditions, e.g., the cortical regions (V1, V2, V4, IT, etc.)/object categories/neural recording vs. behavioral, etc. We have to understand the error consistency under these different dimensions and analyze the insights. More figures/tables regarding this are needed. * For figure 3(c), why do silhouette experiments show a linear correlation between CNNs and humans, compared with “cue conflict” and “edge”? Again, conditional analysis is needed. Also, I am a bit confused about the motivation of using “cue conflict”, “edge”, and “silhouette” for experiments. Normal images should be used as baseline as well. In addition, although the authors argue the CORnet-S model has different processing mechanisms from the human brain given the low score vs. humans, the model does perform well on brain predictivity. The authors should discuss a better unifying way to present the score in the BrainScore benchmark so every valuable aspect is not missing. ======Post-rebuttal======= The authors have addressed most of my concerns (comparison on natural images, more analyses across models and conditions), so I would like to increase my score accordingly. I do think the metric of error consistency is useful and can contribute to further research endeavors in the literature.

Correctness: The method and claims are supported by the results shown in the manuscript.

Clarity: Yes

Relation to Prior Work: Yes

Reproducibility: Yes

Additional Feedback:


Review 4

Summary and Contributions: 1. This paper emphasizes the value of analyzing trial-by-trial response similarity between DNN model and human subjects. The authors proposed a measure that quantifies how consistent the classifications are between the two compared with chance level. 2. Using this criterion, the authors found that DNN models do not make consistent decisions with humans, raising the question of whether DNN models are good models of the brain. However, details of the experiment do not strongly support the findings and the conclusion.

Strengths: The approach of analyzing error consistency is clearly of scientific value. The analysis of confidence internal and bounds are useful and crucial for claiming above-chance consistency. They compared a large number of DNN models with different architecture and performance. It also brings the field to attend the consistency of behaviour rather than aggregate measures. I appreciate that the authors provided all the code for their analysis.

Weaknesses: The main problem of the paper is the use of unnatural stimuli for testing. The mismatch in stimulus distribution can result in all sorts of unexpected behaviour (although interestingly error consistency between models are high). This then leads to contradictions which I describe in the detailed comments. In any case, if the model is not trained at all on the test stimuli, then it is unreasonable to investigate the behaviour of the network. Note that the loss function of the mode during training is only over the realistic ImageNet images. Second, the decision strategy for the DNNs is not clearly described or justified. Does the network choose the top-1 decision, or does it make probabilistic decisions according to the predictive distribution? How does the predictive distribution look like? I would also very much like to see a few examples of predictive distributions in the author response. Third, the presentation of the main message is repetitive. In particular, the value of analysing consistency is brought up multiple times as if the reads will forget this key message of the paper. Section 1.1 tries to summarise the findings in a nutshell while compressing many unclear terminologies and data; this unclarity can be very misleading before giving sufficient details.

Correctness: The analyses of CI and bounds are sound. Presenting stimuli from unseen data distributions is problematic. The above-chance consistency in Figure FS.3 goes unmentioned. It paints a very different picture than Fig 4(a) and can lead to a different conclusion of the paper. Aggregating the classification probability by arithmetic mean may not be optimal, as the distribution of the categories in the training set (ImageNet) may not be consistent with this 16-class test dataset. The authors should check and tell the readers whether this is the case. In line 196, it is not true that better performing networks are better models for vision. Many of the SOTA classifiers are bad models for vision. The cited paper did not optimise the model architecture for performance.

Clarity: The main idea of the paper is simple but was presented in a very repetitive fashion. There are also apparent contradictions.

Relation to Prior Work: The use of consistency measure for studying *behavioural* aspects of DNN models and humans seems novel. However, this is not novel in the broader context in Related Work (61-80). Using DNN unit activations to predict *neural* response can also be seen as an error consistency analysis (between model response and measured response). Representation similarity analysis is also an indirect measure of consistency on the aggregate/system level. The authors should also try to explain why there was no training on the stimuli drawn from the test distribution has as done elsewhere.

Reproducibility: Yes

Additional Feedback: **Contradictions: The authors quoted (175-176) that a good ImageNet performance gives a good performance for transfer learning, but it seems that the networks were not further trained/fine-tuned on the unnatural images, but directly tested on unnatural stimuli. As the authors rightfully pointed out, error analysis is valuable if the performance of the models and humans are high in S.1 (425-426). Still, the performance in Table 1 violates this condition: the accuracies of the models on these untrained stimuli are much worse than human. What's more problematic is that human performance on edge stimuli is highest, while the DNN networks had much higher performance on silhouette images. The higher consistency in the silhouette stimuli in Figure SF.3 seems correlated with the high performance in Table 1. Based on these, the DNNs are already disqualified as models for human responses to these stimuli. ** Suggestions: Since the approach of using error consistency is very nice, this paper would be much more rigorous and compelling by improving other respects. These are my suggestions to improve the quality of the study: In explaining the error overlap, the authors could explicitly mention *correlation* between human and network judgement. This may also help how the bounds in (7) are established (could perhaps also explained by a graphical explanation showing overlapping and non-overlapping proportions of the decisions). Check whether the network gives a confident wrong predictive distribution given these unnatural stimuli. if this is the case, I would not trust that the model is appropriate for these stimuli. Repeat the DNN simulation with natural, ImageNet stimuli, possibly discarding DNNs that are already above human level. Repeat the error consistency analyses by employing different decision rule (probability matching or top-1) for the networks. Or the authors could analyse the effect of a stochastic decision rule. Remove repetitive descriptions of error consistency, for example, in the first paragraphs of sections 2, 3 and 4. Address the difference between Figs SF3 and 4. The statement in the title of section 3.2 is strong and unclear. CORnet-S was tested using realistic stimuli, so this statement here is not warranted by the experiments in the paper that used unnatural stimuli. I also believe the description in 221-224 is not adequate nor objective. ======== update ========== I thank the authors for providing additional experiments in response to the reviews. They address some of my concerns, especially on natural images and consistencies across different stimuli. I raise my score, but emphasise that following concern which was ignored by other reviewers during discussion. The use of untrained stimuli for neural networks. It is true that we use artificial stimuli to probe the biological system, but those artificial stimuli are not completely alient to us: humans see edges and silhouettes through their life time, but the ANNs have NEVER seen them at all. To this extent, the comparisons are very unfair. The authors results on stochastic responses also seem to suggest that DNNs make very confident decisions when seeing these alien images, which could mean those decisions do not make sense.

[Author Response · NeurIPS 2020]

We would like to thank all reviewers for their valuable feedback and we very much appreciate their assessment of our
work as *extremely relevant to the NeurIPS community* and *extremely well written* (**R1**), a *principled evaluation* and
potentially a *highly impactful paper* (**R2**) with *novel* and *very intriguing findings* (**R3**). Furthermore, all reviewers were
confident that our work can be reproduced, and pointed out how it could *encourage the field to explore a wider space of*
*architectures and training schemes* (**R2**) and *attend the consistency of behaviour rather than aggregate measures* (**R4**).

**R1**, **R2**, **R3**, **R4**: *Discussion of Brain-Score / CORnet is overly critical. Find way of unifying*
*benchmarks.* We apologise for our overly critical presentation of Brain-Score and CORnet. Together
with CORnet/Brain-Score authors Kubilius and Schrimpf we re-phrased numerous unfair or misleading
statements and now have a balanced manuscript we and K&S all agree upon. K&S believe error
consistency to be an important behavioural metric and want to include it on Brain-Score.

**R1**: *CNNs are not trained on stimuli* / **R2**: *Repeat experiment on dataset where CNNs and humans*
*have similar performance* / **R3**, **R4**: *Repeat experiment with natural ImageNet images as baseline.*
We now include standard ImageNet images where human and pre-trained CNN accuracies are both
very high and similar ($.960 \pm .036\%$). New results included in the paper (shown on the right)
complement previous findings. Thanks for this excellent suggestion which makes the paper stronger!

**R1**: *Clarify term "strategy".* We now clarify the difference between "high-level strategy" and "decision rule" (for
decision rule: following the terminology from "Shortcut learning in deep neural networks", GEIRHOS ET AL, 2020).

**R1**: *Only CNNs trained on ImageNet were used* / **R2**: *Include models trained on Stylized-ImageNet.* Again an
excellent suggestion which we have incorporated into our final manuscript. We analyzed three CNNs with different
degrees of stylized training data. Model shape bias predicts human-CNN error consistency for cue conflict stimuli,
indicating that networks basing their decisions on object shape (rather than texture) make more human-like errors:

| model shape bias (%) | 20.5 | 21.4 | 34.7 | 81.4 |
|---|---|---|---|---|
| human-CNN consistency ($\kappa$) | .066 | .068 | .098 | .195 |

**R2**: *Does kappa really disentangle error consistency from accuracy?* In Figure 2b, $\kappa$ and $c_{exp}$ are not correlated
(r=-0.00015, $p > 0.05$); a simulation (see right plot) confirms: $\kappa$ and accuracy are not correlated for independent
decision makers (r=-0.004, $p > 0.05$). For *dependent* observers, any pattern is possible: zero correlation (Figures
3a, 3b), positive correlation (Figure 3c) and even negative correlation (simulated toy experiment, bottom figure on
the right). Thus crucially, there is *no* correlation between consistency ($\kappa$) and accuracy for independent observers
whilst for dependent (consistent) observers correlations are possible but they are a property of the decision makers,
not the analysis. We now discuss this point in the main paper and show the simulations in the appendix.

**R3**: *A closer analysis of error differences would be helpful. / Detailed comparison to Brain-Score.*
Another nice suggestion! We now visualize striking error differences between CNNs and humans for all
experiments (original ImageNet images, cue conflict, silhouette, edges) and discuss potential underlying causes.
Example visualized below. Top row: "Hard" images for CNNs (correctly classified by all humans but not by *any*
CNN). Bottom row: "Hard" images for humans (`bear`, `bear`, `bird`, `oven`). Additionally, we now plot confusion
matrices to analyse category-level errors. Concerning comparison to Brain-Score conditions V1, V2, V4, IT,
behaviour: This is already done in the appendix (SF.7, SF.8, SF.9); now linked & discussed more prominently.

**R4**: *The main problem of the paper is the use of unnatural stimuli for testing.* First we now include
"natural" ImageNet images leading to the very same conclusions as with "unnatural" stimuli (see
above). Second, we strongly believe in the value of investigating model behaviour with controlled
"unnatural" stimuli: Significant progress in neuroscience—e.g. discovering receptive fields of simple
and complex cells—was made using "unnatural" bar-like stimuli. In deep learning adversarial
examples and texture bias were discovered by testing models on (unnatural) images different than
the training data. Clearly we can learn a lot about the inner workings of a system by probing it
with *appropriate* artificial stimuli ("In praise of artifice", RUST & MOVSHON, 2005; "In praise of artifice reloaded",
MARTINEZ-GARCIA ET AL, 2019); we now state this motivation more explicitly.

**R4**: *Aggregating the classification probability by arithmetic mean may not be optimal.* We have now included
a principled derivation showing that the arithmetic mean is, perhaps counterintuitively, optimal. Essentially, the
aggregation can be derived by calculating the posterior distribution $p(c|x)$ of a discriminatively trained CNN under a
new prior chosen at test time (here: $\frac{1}{16}$ over coarse classes); resulting in decision $C|x = \text{argmax}_C \sum_{c \in C} \frac{1}{|C|} p(c|x)$.

**R4**: *Other suggestions.* (1) We now state why we did not include a comparison of $\kappa$ to the (Pearson) correlation
coefficient since the literature rejects correlation as a measure of agreement (HUNT 1986, WATSON & PETRIE, 2010).
(2) We used deterministic decisions throughout the paper. When a proportion of decisions are stochastically sampled
from the softmax instead, consistency between two CNNs decreases slightly (see plot on the right).



[Meta-Review · NeurIPS 2020]

The rebuttal and additional analyses seemed to satisfy most of the reviewers, so thanks to the authors for their thorough and responsive reply. Overall this seems like important, well-presented work, and hence I'm recommending acceptance. That said, there was a suggestion to apply the analysis of figure 4 to natural images (it seems like from the initial analysis included in the rebuttal that the results shouldn't be surprising). I sympathize with the argument that artificial stimuli are often used in neuroscience, but in this case I think performing this analysis on natural images would provide a nice sanity check, and I think many NeurIPS readers would also be curious. I don't think that lack of this analysis is enough to reject the paper, but I'd like to request authors include this for the final version.